# Understanding the Behavioral Intentions about Holidays in the Shadow of the COVID-19 Pandemic: Application of Protection Motivation Theory

**DOI:** 10.3390/healthcare10091623

**Published:** 2022-08-25

**Authors:** Kevser Çınar, Saadet Zafer Kavacık, Ferdi Bişkin, Muhsin Çınar

**Affiliations:** 1Faculty of Tourism, Necmettin Erbakan University, Köyceğiz Yerleşkesi, Köyceğiz Mah. Demeç Sk. No: 42, Meram, 42140 Konya, Turkey; 2Institute of Social Sciences, Nigde Omer Halisdemir University, Merkez Yerleşke Bor Yolu Üzeri, 51240 Niğde, Turkey

**Keywords:** protection motivation theory, fear, hope, emotional response, behavioral intentions, holiday intention, holiday avoidance, domestic tourists, COVID-19

## Abstract

The research aims to investigate the emotional response and protective behaviors of domestic tourists’ post-pandemic period and their holiday intentions or holiday avoidance behaviors. For this reason, understanding tourist behavior during and after significant tourism crises is critical for the recovery of the tourism industry. To achieve this aim, first, we examine the effects of perceived vulnerability and perceived severity factors in the threat appraisal of domestic tourists, the effects of the response efficacy, response cost, and self-efficacy factors in the coping appraisal, and the effects of fear and hope factors as the anticipatory emotion responses regarding protection motivation. Second, we measure the effect of protection motivation on the factors of taking a vacation and avoiding a vacation, which constitute behavioral intention. This study applies the health-related protection motivation theory to explore how domestic tourists’ behavioral intentions are influenced by Coronavirus disease (COVID-19) in the post-pandemic period. The study includes citizens residing in Türkiye who have had at least one-holiday experience in the last five years. Online questionnaire surveys were administered to 1391 domestic tourists. In the research, in addition to testing the validity and reliability of the scales, simple linear regression analysis was used to test the model based on the hypotheses experimentally. The results show that factors have internal consistency reliability, convergent validity, and discriminant validity. Response cost and hope variables are ineffective in predicting the protection motivation, and all other effect sizes (*f*^2^) are positive. All hypotheses have been supported. However, the response cost (β = −0.029, *p* > 0.05) has no effect on protection motivation, thus only one is rejected. As a result, domestic tourists would like to maintain the assurance of their health and safety during a holiday. An integrated model with protection motivation theory and different theories as theory of planned behavior should be implemented. As a result, this will provide a more comprehensive understanding of the complexity involved in the sustainable behavioral intentions in the post-COVID era.

The abstract of this research was accepted to be presented as an oral presentation in terms of the 8th Annual Conference of EATSA (Euro–Asia Tourism Studies Association) on the 4th of July 2022 and will be published in the conference proceedings.

## 1. Introduction

COVID-19 is the pandemic of the century that poses a significant threat to the world and has permanently changed people’s lives. It has unleashed unprecedented contagion fears among tourists and its extraordinary circumstances have increased fear, anxiety, and other negative emotions [1,2,3], leaving people haunted by terrible remembrances and hopelessly forced to live with that pandemic [4]. Understanding tourist behavior during and after significant tourism crises is critical for the recovery of the tourism industry. Most studies regarding post-pandemic holiday behavior have centered on the impact of the industry [5] or tourism demand [6] without identifying the psychological attributes associated with these behavioral changes. For example, studies have shown that Ebola caused a considerable drop in tourism demand in Africa in the year following its outbreak [7]. After the SARS epidemic, Chinese tourists shifted their holiday behavior to maintain social distance, such as preferring remote places, limiting group travel, and avoiding interaction with others while traveling [8]. Several public health studies have shown that risk avoidance behaviors positively accompany fear during or post-disease outbreaks (e.g., [9]). Therefore, understanding individuals’ protection motivations regarding the pandemic is crucial to assist in the review of the tourism industry after the outbreak.

The pandemic has had a significant impact on the tourism industry, with significant impacts on illness, death, and unemployment [10]. A variety of risk factors demonstrate the significant predictive power of individuals’ intentions to postpone international tourism plans amid COVID-19 [11]. When a health crisis triggers tourists fear, they might prefer not to travel to decrease possible risks [7]. Adopting protection motivation theory (PMT), studies regarding holiday risks have justified that protection motivation of tourists is a strong indicator of their behavior avoidance (e.g., [12]) or acceptance of preventive behaviors in tourism (e.g., [13]). Therefore, protection motivation may considerably impact tourists’ holiday avoidance and intention to travel cautiously during the post-pandemic period. Furthermore, resilience can considerably enhance an individual’s adaptive behavior (e.g., planning, protection) in the face of threats [14].

In this context, PMT has received scientific attention to reveal self-restraint and precautionary holiday behaviors during the pandemic [15]. The theory proposes protective behaviors that arise from a circumstantial threat, such as a health crisis, coping, and threat appraisals [16].

PMT has been used to study health-related behaviors, such as reducing alcohol consumption (e.g., [17]), leading a healthy lifestyle (e.g., [18]), and protecting children from myopia [19] and disease prevention (e.g., [20]). However, this theory has been applied by only a few researchers in tourism. For instance, Sönmez and Graefe [21] concluded that perceived risks and safety are the important indicators of avoidance behavior rather than visiting a destination. They found that “if the risk is associated with international travel, leisure travelers are well placed to adopt protective measures” (p. 176). Moreover, the authors [22] investigated international tourists’ protective behavior against health threats (e.g., rabies) through full acceptance of PMT. The results suggested that Australian travelers’ protective motives, such as threat and coping appraisals, may incline the intention to protect, which is eventually reflected in actual behavior. Ouintal et al. [23] reported that the differential impacts risk and uncertainty have on travel decision-making were explored by examining the constructs’ influence on the antecedents of intentions to visit Australia using the theory of planned behavior. The results suggested that both perceived risk and perceived uncertainty negatively influenced people’s attitudes toward visiting Australia.

However, no studies have investigated the relationship between PMT constructs and the behavioral intentions of domestic tourists about a holiday in Türkiye. If psychological factors that predict the behavioral intentions during the COVID-19 pandemic are revealed, the findings will contribute to more persuasive campaigns to encourage people to travel in Türkiye. We are particularly interested in the relationship between health-related theory PMT and behavioral intentions. Since most studies regarding post-pandemic holiday behavior have centered on the impact of the industry [5] or tourism demand [6], there is a serious gap in the literature to understand the psychological attributes associated with the behavioral changes. Therefore, the aim of this research is to investigate the emotional response and protective behaviors of domestic tourists’ post-pandemic period and their holiday intentions or holiday avoidance behaviors. To achieve this purpose, first, we examine the effects of perceived vulnerability and perceived severity factors in the threat appraisal of domestic tourists, the effects of response efficacy, response cost, and self-efficacy factors in the coping appraisal, and the effects of fear and hope factors as the anticipatory emotion responses regarding protection motivation. Second, we measure the effect of protection motivation on the factors of taking a vacation and avoiding a vacation, which constitute behavioral intention.

Moreover, suggesting a structural model in accordance with the results of these effects is a crucial output of the research. Therefore, this study will assist us in understanding how the perceptions of domestic tourists regarding COVID-19 have changed and how the findings can be used to improve the image and marketing strategies of destinations. In addition, we will explore new market opportunities and target today’s diverse consumer cohort. The key questions underpinning the research are (1) Do the emotional response and protective behaviors of domestic tourists affect the protection motivation in the post-pandemic period? (2) Does protection motivation have an effect on behavioral intentions? Therefore, we scrutinized the relationship between emotions and behavioral intentions by examining the subdimensions of the concepts, their interactions, and their combined impact on tourists’ protective behavior in the post-COVID era through the lens of PMT.

## 2. Literature Review

### 2.1. Protection Motivation Theory and Its Variables

An individual’s intention to perform a particular behavior is the main stimulator [24]. Several psychosocial theories have been established to predict, explain, and alter health behaviors. These theories fall into two broad categories, commonly referred to as social cognition models. The term “models of social cognition” refers to a group of similar theories that specify a small number of cognitive and affective factors, such as beliefs and attitudes as direct behavioral determinants. The most popular models among health behavior researchers in recent years are the health belief model, PMT, self-efficacy theory, the theory of reasoned action, and the theory of planned behavior. As a socio–cognitive model, PMT is commonly used by researchers to examine and understand various health-related behaviors [25] and to forecast people’s motivation to practice protective behaviors [26].

The PMT was first developed to explain the effects of fear on health-related attitudes and behaviors [16]. Knowledge regarding a health threat initiates the cognitive mediation process. These processes evaluate the congruent response(s) or the maladaptive response(s). For example, in accordance with the cognitive evaluation process, a person may continue to smoke or try to quit [27]. PMT is organized through two cognitive mediation processes, the threat and coping appraisal. Threat and coping appraisals were used to establish protection motivation as a mediator variable [28].

The theory of protection motivation is used extensively in health-related topics, such as health promotion, the prevention of illness and injury, as well as political issues, environmental concerns, and social media campaigns [28]. The PMT suggests that a person’s commitment or determination to engage in health-protective behaviors relies on four fundamental insights or perceptions: The severity of the threat, the personal vulnerability to the risks, the response efficacy of the risk-reduction behavior, and the self-efficacy in performing the risk-reducing behavior [29]. Self-efficacy, a central concept in the social learning theory proposed by Bandura [30], is recognized as an essential determinant of behavior. It indicates the beliefs regarding one’s ability to perform a behavior successfully.

PMT has also been applied in tourism to examine tourists’ protection behaviors against health crises in tourism movements [31]. Ruan et al. [12] tried to determine the tourists’ behavioral intentions under the threat of air pollution in China. They found that the threat appraisal factor, perceived severity, had the most significant impact. Sönmez and Graefe [21] studied the issue of determining future travel behaviors based on past travel experiences, risk, and safety perceptions. As a result, they found that perceived risk and safety factors were stronger predictors of avoiding the area than planning to visit. Lu and Wei [32] studied the perceived risk of Chinese people to overcrowding and the adoption of precautionary measures during holidays. They found that risk perception had a significant impact on adopting measures. Fisher et al. [13] tried to determine the effectiveness of the PMT in predicting cruise passengers’ handwashing intentions in COVID-19 cases. They found that the coping appraisal process significantly affected the handwashing intention, and the cost was the most influential sub-factor. The new normal amid COVID-19 changed conventional lifestyles, travels, and interactions, reversing traditional behavioral intentions and giving life a new meaning and value. Although PMT has been used to predict and understand a wide variety of health-related behaviors, there is limited research in terms of COVID-19 prevention [33].

Wang et al. [22] extensively studied individuals’ health behaviors during travel using the PMT model. They concluded that threat and coping appraisals could increase individuals’ protection motivation, affecting their actual behavior. Threat appraisal measures maladaptive responses [34]. Individuals’ perceptions of the severity of the threat and their vulnerability to this threat seem to prevent maladaptive responses [35]. Perceived vulnerability is an individual’s assessment of how personally susceptible they are to the threat posed, while perceived severity is expressed as the individual’s evaluation of how serious the threat will be in his/her own life [25]. In the case of COVID-19, the perception of threat is related to individuals’ perceptions of the undesired consequences of COVID-19 that may result in negative effects on their lives [36].

Coping appraisal focuses on the adaptive response. It evaluates one’s ability to cope with and avoid health-threatening hazards [34]. PMT does not presume that people provide reasonable judgements. Any cognitive evaluation process can be influenced by pragmatic judgement about its significance. During the COVID-19 pandemic, messages that solicit fear may be more efficient in encouraging protective behaviors among those who already perform them [37]. Response cost, response efficacy, and self-efficacy variables, which represent coping assessment, are used in the research. Response cost is related to the belief regarding how much it will cost the individual to perform the proposed behavior. However, the effect of response cost on behavioral intentions remains an unanswered question. Although the literature generally agrees that it should act as a behavior inhibitor, it also reports mixed results about its significance in explaining human behavior or omits the problem [18]. In contrast, response efficacy is related to the belief that the proposed behavior will effectively reduce the threat to the individual [25]. In accordance with [28], self-efficacy has only been studied as a separate component in the theory of protection motivation among the other theories discussed by Weinstein [38]. Individuals with higher self-efficacy believe that they can perform a specific behavior or set of behaviors [39]. Self-efficacy requires not only skills, but also a strong belief that one can exert control over one’s motivations and behaviors [40]. Therefore, the first set of research hypotheses has been proposed:

**Hypotheses** **H1.**
*Individuals’ perceived vulnerability increases their protection motivation.*


**Hypotheses** **H2.**
*Individuals’ perceived severity increases their protection motivation.*


**Hypotheses** **H3.**
*Individuals’ response efficacy increases their protection motivation.*


**Hypotheses** **H4.**
*Individuals’ response cost increases their protection motivation.*


**Hypotheses** **H5.**
*Individuals’ self-efficacy increases their protection motivation.*


### 2.2. Anticipatory Emotions and Protection Motivation Theory

The classical theory of primary emotion proposed by Ekman [41] ascertains emotions as natural and widely shared intrinsic mental conditions (e.g., happiness, anger, and sadness) [42]. As stated by PMT, a pandemic case unavoidably requires an assessment of the severity and vulnerability of the risks involved and convenient coping appraisals. The unbalanced coincidence of the high risk of reversal and inaccessible resources triggers emotional and affective responses to the condition, generating a case of tension [43]. A stimulating acquaintance of people during a crisis is perplexing due to various negative sentiments [44]. For example, fear of COVID-19 was prevalent as the initial COVID-19 outbreak was described as undisclosed but disastrous; fear later turned into depression [3]. Emotional fear, among other sentiments, has been highlighted as a natural efficient reaction derived from the threat appraisal of the pandemic [2,43]. In the light of tourism research, sentiments have attracted consideration as a critical determinant in predicting travel behavior during the COVID-19 pandemic [43]. For example, fear of travel is aligned with protective travel behavior and decreases the desire to travel [43]. Occasionally, people fear or hope that something undesirable may/may not happen in the future. They imagine actions they can take to avoid impending danger and visualize their release or enjoyment when the negative outcome is not realized. These affective reactions are anticipatory emotions because they are currently experienced due to something that could happen in the future [45]. Fear-inducing communication significantly affects the choice of behavior [28].

In the extant tourism literature, protective holiday behaviors during a pandemic are primarily attributed to threat appraisals and risk perceptions [46], as well as emotional responses that lay open to these threats [43]. Although the function of negative sentiments, such as fear and anxiety, in influencing willingness to travel has been highlighted [2], the assumption of this direct causality is criticized for oversimplification fallacy. The role of fear in explaining responses to threats has also been reinforced by adopting hyper defensive or precautionary strategies [29]. In this context, few critical elements of the underlying intervening mechanism may have been overlooked [47].

Kim et al. [2] explained hope as the sentiment of desiring a favorable consequence or release from the negative effects that COVID-19 generates on the person and society. Moreover, the authors interpret fear as an unpleasant sentiment induced by the negative effects of COVID-19 on the person and society. Furthermore, the mixed emotional reaction during COVID-19 develops under circumstances that promote hope: A considerable amount of threat associated with paramount instability.

Baumgartner et al. [45] elaborated on the critical disparities between the two kinds of prospective sentiments by equalizing anticipatory emotions’ positive and negative forms with hope and fear. They defined anticipatory sentiments as any emotional response to cognitive models of upcoming cases and intentions. The PMT does not assume that individuals make rational decisions. A heuristic judgment can bias each cognitive appraisal process regarding its importance. Given the magnitude of the COVID-19 pandemic, fear appeal messages can be more effective in reinforcing protective behaviors among individuals already practicing them [37].

This study builds on recent suggestions (e.g., [48]) to distinguish the future-oriented effective responses in terms of anticipatory emotions. These forward-looking effective responses, such as anticipatory emotions, which COVID-19 is shaking out and reshaping, provide a suitable complement to guide holiday decisions and explain and predict behavioral intentions [49]. To the best of the authors’ knowledge, no study has looked carefully at the effects of COVID-19 on acquired and subsequent vacation behavior in terms of PMT and anticipatory emotions. An individual may feel an emotion (i.e., hope or fear) at the prospect of a desirable or undesirable future event. These effective responses are anticipatory emotions that individuals undergo in the present as a response to something that could happen in the future. The missing link between these anticipatory emotions and holiday propensity in the context of the pandemic and protection motivation theory needs to be demonstrated empirically to improve the understanding of post-COVID-19 holiday behavioral consideration. To address this gap, the present study attempts to improve the understanding of protective travel behaviors during the pandemic by examining the interaction of anticipated emotions and protective behavioral intentions in domestic tourists’ holiday decision-making process. Therefore, we propose this set of research hypotheses:

**Hypotheses** **H6.**
*Fear has a positive influence on protection motivation.*


**Hypotheses** **H7.**
*Hope has a positive influence on protection motivation.*


### 2.3. Protection Motivation Theory and Behavioral Intentions

Several types of research have been carried out to explain the impact of COVID-19 on people’s holiday intentions [43,50,51]. For example, Gupta et al. [50] confirmed an important positive correlation between the perceived risk of traveling and perceived severity and vulnerability to COVID-19 on the one hand and travel avoidance intention on the other hand. However, they concluded a reverse relationship between self-efficacy and travel avoidance intention. Zheng et al. [43] found that threat severity and vulnerability can increase “travel fear,” which drives protection motivation and protective post-pandemic travel behavior. Moreover, the results showed that “travel fear” can trigger different handling tactics. Similarly, Peric et al. [52] found that Serbian tourists’ risk perceptions had a negative impact on their holiday intentions during the pandemic, while Boto-Garcia and Leoni [53] reported that people subjected to COVID-19 were moderately reluctant to go on holiday. Liu et al. [54], who contributed to the growing literature in this area, found that perceptions of COVID-19 negatively affected overseas travel to China, while risk tolerance negatively mitigated overseas post-COVID travel intentions in China. Xie et al. [55] also found an important moderating effect of empathy and perceived waiting time on the relationship between risk messages and holiday intentions after the pandemic.

PMT includes threat and coping appraisal variables, explaining avoidance and the intention to initiate in protective behaviors [56]. In most studies related to PMT, perceived severity, perceived vulnerability, response efficacy, self-efficacy, and response cost variables are mainly used [35]. Response efficacy and self-efficacy enhance the probability that a person will display efficient protective behaviors. The threat appraisal process, which evaluates the perceived severity and vulnerability of the threat, explains the potential harm of an individual to him- or herself or others when he/she takes no action. The difference between feelings about the severity of the threat and the perceived vulnerability increases the risk perception and the possibility of selecting adaptive behaviors, such as the implementation of COVID-19 protective behaviors [57]. A high level of perceived severity and perceived vulnerability related to the threat appraisal, together with a high level of self-efficacy and responsive efficacy, strengthens the intention to act in a protective behavior [57].

Protection motivation is a mediating variable that initiates, maintains, and directs healthy behavior. Protection motivation is a result of threat and coping appraisal. It facilitates the adoption of adaptive behaviors and can best be measured by behavioral intentions [25,28]. Baumgartner et al. [45] proposed that expectant and anticipatory emotions promote the formation of intentions to engage in behaviors aimed at achieving or avoiding future events toward which these emotions are directed. The PMT does not assume that individuals make rational decisions. Each of the cognitive appraisal processes can be biased by heuristic judgment regarding its importance. In addition, research has shown that how tourists perceive risk affected their post-disaster travel behaviors (e.g., [58]) and their motivation (e.g., [59]). Although pandemics can cause psychological distress to tourists, research has rarely analyzed the psychological reactions and coping mechanisms of individuals associated with post-pandemic holiday scenarios. The research model is shown in Figure 1. Then, we propose these sets of research hypotheses:

**Hypotheses** **H8.**
*Protection motivation has a positive influence on holiday intention.*


**Hypotheses** **H9.**
*Protection motivation has a positive influence on holiday avoidance.*


## 3. Method

### 3.1. Construct Measures

In the research, a quantitative research approach was used to test the model based on hypotheses. The participants were asked to answer demographic information, such as gender, age, education level, income level, and field of study. At the same time, they were asked to share information regarding the number of domestic and international holidays in the last five years and the history of COVID-19. Previous studies in the literature were used to measure the constructs that make up the model together with demographic questions. Measures for the suggested research framework were adapted from studies dealing with PMT, hope, fear, and behavioral intentions. The source and items used in the research are shown in Table 1. A 5-point Likert structure, ranging from Strongly Disagree (1) to Strongly Agree (5), was used to measure perceived vulnerability and perceived severity dimensions consisting of three statements each. The response efficacy dimension consisted of five statements, the self-efficacy and protection motivation dimensions consisted of four statements, and a 5-point Likert structure, ranging from Strongly Disagree (1) to Strongly Agree (5), was used to measure the statements. The fear dimension consisted of three statements, and a 5-point Likert structure, ranging from Not at all (1) to Extremely (5), was used in the scale of the statements. A 5-point Likert structure, ranging from Strongly Disagree (1) to Strongly Agree (5), was used in the measurement of the response cost dimension, which consisted of seven statements. The hope dimension consisted of three items, and a 5-point Likert structure, ranging from None (1) to Extremely (5), was used to measure the dimension. A 5-point Likert structure, ranging from Strongly Disagree (1) to Strongly Agree (5), was used in the measurement of the holiday intention dimension, which consisted of five statements. The holiday avoidance dimension consisted of seven statements and was measured with a 5-point Likert structure, ranging from Strongly Disagree (1) to Strongly Agree (5).

Ethical permission to conduct the research was obtained by the Necmettin Erbakan University Social and Human Sciences Scientific Research Ethics Committee (Decision number 2021/575 on 10 December 2021). Researchers were informed about the use of the scales, and permission was obtained. The semantic equivalence of the questionnaire was provided by the translation and back translation approach. Two translators independently translated the English items into Turkish. Afterward, a panel discussion was held with experts to discuss and review the differences between the two versions of the translations.

### 3.2. Data Collection

The study included citizens residing in Türkiye who had at least one-holiday experience in the last five years. Since the participants were asked to have had a recent vacation experience, the duration was limited to five years. The survey was created and distributed through a well-known online survey service company (www.surveey.com; accessed on 1 January 2022), which has recently been widely used in scientific studies in Türkiye [62]. Because of the pandemic and hygiene conditions, the survey was conducted online. The factors that effectively determine the sampling method are the purpose of the research, the definition of the population, the data collection technique, the design of the research, the financial source, the time, and the possibility of control. Sampling methods are generally classified as random and non-random sampling methods. What is meant to be emphasized with the word random here is the probability that each unit in the population can be part of the sample and therefore, the sample can represent the population correctly [63]. In the study, convenience sampling and snowball sampling, which are non-random sampling methods, were used together. The reason for using convenience sampling is due to the fact that since the research population consisted of citizens residing in Türkiye who had at least one-holiday experience in the last five years, preliminary numerical data about the population could not be obtained. Therefore, the research was carried out on individuals who were accessible. The reason for using snowball sampling is to use the time efficiently by reaching other citizens through the people within the sample.

Prior to the primary implementation, a pilot test of the online questionnaire was conducted with 100 participants to check the suitability of the questionnaire. The primary implementation was carried out between January and February 2022. After the final questionnaire form was created, the link was shared in social media channels with 250 people in total, including the researchers. These groups included academics, students, and members of the civilian population. The people to whom the survey link was shared were asked not only to answer the survey, but also to share it with people who might be interested in the survey and in their own groups. By the end of February 2022, 1547 people completed the survey. Among these data, 156 were excluded from the research due to missing values, and the analysis process was started with a total of 1391 surveys.

### 3.3. Data Analysis

The IBM SPSS Statistics 28 program was used in the research to determine descriptive statistics, simple linear regression analysis, and to measure the mediation effect with PROCESS v4.0. To determine the surveys’ reliability values (Cronbach alpha (CA) and composite reliability (CR)), values of convergent validity (average variance extracted (AVE)), discriminant validity values (root square of AVE and Heterotrait–Monotrait ratio of correlations (HTMT)), as well as the power of the statistical relationship between the variables [64], LISREL 8.80, and Microsoft Excel programs were used. In addition, confirmatory factor analysis was conducted to determine the validity and reliability values of the scales regarding perceived vulnerability, perceived severity, response efficacy, response cost, self-efficacy, hope, fear, protection motivation, holiday intention, and holiday avoidance.

## 4. Results

### 4.1. Profile of Respondents

Detailed information about the profile of the participants is shown in Table 2.

In total, 54.3% of the participants were female, and 45.7% were male. Most of the participants were in the 34–41 year age range (30.4%) or in the 26–33 year age range (25.4%). In addition, 58.3% of the participants were undergraduates, 35.6% were in the 5001 ₺–10.000 ₺ income range, and 42.3% were working in the private sector. Moreover, 31.6% had a holiday at least once, while 19.1% stated that they had had four or more holiday experiences in the last five years. In accordance with their answers about COVID-19 stories, 65.7% of the participants stated that they had COVID-19 before, and 82.5% stated they had a COVID-19 vaccine.

### 4.2. Descriptive Statistics

Table 3 shows the arithmetic mean, standard deviation, skewness–kurtosis values, and factor loads obtained due to confirmatory factor analysis of the scale. One of the necessary elements to perform confirmatory factor analysis is that the data show a normal distribution. Different reference intervals are considered a basis for evaluating the normal distribution in accordance with the skewness. Hair et al. [65] stated that the range of ±1.0 and Tabachnick and Fidell [66] stated that the range of ±1.5 should be considered a reference. Meanwhile, George and Mallery [67] stated that a kurtosis value between ±1.0 is acceptable for most psychometric purposes, and a value between ±2.0 is also acceptable in most cases, depending on the specific application. The skewness–kurtosis values in Table 3 varied between ±1.0, and the data were normally distributed.

To determine factor structures for construct validity, confirmatory factor analysis was performed on all dimensions in the Lisrel 8.80 program. There are parameters related to the variables that need to be removed from the model, and the error variances that are deemed appropriate need to be added to the variables. Making changes in the model, provided that they do not go beyond the purpose of the model and are based on a theoretical logic, can lead to better fit values [68]. Items with an error variance close to one hundred and factor items with a t value below 1.96 and a factor load less than 0.50 were excluded from the analysis. The lowest factor load was 0.52, while the highest factor load was 0.96. In terms of fit values, there was a fit between the factors and the observed data, and the paths from the factors to the propositions had good fit indices. A total of 10 items were excluded from the analysis, and the analysis continued with 34 items and 10 dimensions. Excluded items included “Measures that can be taken to stop tourists from being infected by COVID-19 are adequate”, “Preventive measures to stop tourists from being infected by COVID-19 are adequate”, “The price of disinfectants is high”, “Disinfecting objects and places suspected of COVID-19 is time-consuming”, “It is hard for me to stay at home and not be in crowded places”, “Washing hands frequently with soap and water can hurt the skin of my hands”, “I intend to go on holidays during 2022”, “I will have no problem with using planes, buses or trains as they will be safe again soon”, “This year I would rather look for holiday possibilities within my own country”, and “Once the problems are over, I will travel extensively to make up for lost time”.

The arithmetic means and standard deviations of the answers provided by the participants to each statement were obtained. Scoring range = (upper value − lower value)/number of value was calculated with the formula, and the score interval for arithmetic averages was calculated as 0.80 ((5 − 1)/5 = 4/5 = 0.80) [69]. Evaluation ranges and corresponding items are shown in Table 4.

When Table 5 is examined, it can be seen that the participants generally agreed with the statements on the scale at a medium level. In addition, there was a negative level of participation in the dimensions of the response cost (2.4299 ± 0.93993) and hope (2.5037 ± 0.98370), while a positive level of participation was seen in the dimensions of perceived severity (3.6336 ± 1.02815), response efficacy (3.4682 ± 0.90593), and protection motivation (3.5286 ± 0.91704).

### 4.3. Measurement Model Evaluation

The research tested the internal consistency reliability, convergent validity, and discriminant validity of the constructs.

In accordance with the findings, the Cronbach alpha (CA) value ranged from 0.64 to 0.91, and the composite reliability (CR) ranged from 0.79 to 0.93. The average variance extracted (AVE) value for each structure was between 0.51 and 0.82, higher than the 0.50 AVE threshold [70]. If the CA value is between 0.60 and 0.80, the scale is quite reliable [71]. Therefore, PV was accepted. The discriminant validity was examined by comparing the square root of AVE with the latent variable correlations and using the heterotrait–monotrait ratio inference (HTMT inference). The diagonal elements of the correlation matrix represent the square root of the AVE (values in bold) and the off-diagonal elements represent the correlation values between the dimensions. As seen in Table 6, the square root of the AVE values calculated for each dimension was greater than the correlation values with the other sub-dimensions.

Table 7 shows the HTMT values prepared in accordance with the correlation values between the items in each dimension. Although the HTMT_85_ and HTMT_90_ criteria are described as more stringent evaluation criteria, the HTMT_inference_ value reliably determines the discriminant validity [72]. When the data in Table 7 were examined, the comparisons of self-efficacy–reaction adequacy (0.886) and holiday avoidance–going on vacation (0.856) indicated in the gray areas did not meet the HTMT_85_ criterion. The comparisons of protection motivation–response efficacy (0.910) and protection motivation–self-efficacy (0.964) did not meet the HTMT_90_ criterion. However, HTMT_inference_ was detected as 0.655, and the discriminant validity between dimensions is provided.

### 4.4. Estimated Structural Model

In the study, the effect size (*f*^2^), coefficient of determination (R^2^), path coefficients (β), t value, significance value, and standard error values were used to estimate the structural model. The *f*^2^ values represent the contribution of the predictor variables to the dependent variables, which can be defined as small (*f*^2^ > 0.02), medium (*f*^2^ > 0.15), and large effect sizes (*f*^2^ > 0.35) [61].

The findings in Table 8 reveal that the response cost and hope variables were ineffective in predicting the protection motivation, and all other effect sizes (*f*^2^) were positive. Perceived severity (*f*^2^ = 0.70), response efficacy (*f*^2^ = 1.21), and self-efficacy (*f*^2^ = 1.98) had a large effect, while perceived vulnerability (*f*^2^ = 0.33) had a medium effect. In addition, fear (*f*^2^ = 0.06) had a small effect, protection motivation had a medium effect on holiday intention (*f*^2^ = 0.18), but had a small effect on holiday avoidance (*f*^2^ = 0.14). Looking at the R^2^ values, the perceived vulnerability dimension explained the protection motivation with a percentage of 25% (R^2^ = 0.249); perceived severity, 41% (R^2^ = 0.413); response efficacy, 55% (R^2^ = 0.548); self-efficacy, 66% (R^2^ = 0.664); and fear, 6% (R^2^ = 0.055). However, it could not explain the dimensions of the response cost (R^2^ = 0.001) and hope (R^2^ = 0.003). These findings are similar to the effect size findings.

The results of the research hypotheses are also presented in Table 8. The findings show that the protection motivation was affected significantly by the perceived vulnerability (β = 0.499, *p* < 0.001); perceived severity (β = 0.643, *p* < 0.001); response efficacy (β = 0.740, *p* < 0.001); self-efficacy (β = 0.815, *p* < 0.001); fear (β = 0.235, *p* < 0.001); and hope (β = 0.059, *p* < 0.05) variables. Therefore, all hypotheses H_1_, H_2_, H_3_, H_5_, H_6_, H_7_, H_8_, and H_9_ were supported. However, the response cost (β = −0.029, *p* > 0.05) had no effect on protection motivation, thus H_4_ was rejected. In terms of the t values, all t values were higher than 1.96, except for the response cost (t = −1.073 <1.96), and they were significant with a margin of error of 0.05. The estimated structural model is shown in Figure 2.

In Table 9, the total impact, direct impact, and indirect impact results, which are predicted to be among the variables, are provided. Preacher and Hayes [73] stated that the lower limit (BootLLCI) and upper limit (BootULCI) values of bootstrap confidence interval statistics should be examined to determine the significance of the mediation effects in the model. If both of these values are below or above zero, it indicates that the mediating effect is significant. As can be seen in Table 9, the BootLLCI and BootULCI values were above zero, except for four of the 14 indirect effects. It has been determined that there was no intermediary effect between the response cost and holiday intention, hope and holiday intention, response cost and holiday avoidance, and hope and holiday avoidance. Moreover, the independent variable must have a significant effect on the dependent variable as a prerequisite for examining the mediation effects [74]. For this reason, it is a normal result in terms of the theoretical perspective, as the response cost did not have an effect on the protection motivation, and the protection motivation variable was not a mediator between the response cost and the holiday intention and holiday avoidance variables. Meanwhile, the hope variable had a significant effect on protection motivation. It was found that the protection motivation did not have a mediating effect on the holiday intention and holiday avoidance in terms of the hope variable. When all the data related to the hope variable were evaluated together, the R^2^, β, and t values were low, and this result was not a surprise.

## 5. Discussion

The primary purpose of this research was to investigate the emotional response and protective behaviors of domestic tourists in the post-pandemic period, as well as their holiday behavioral intentions or avoidance behaviors. The aim was to examine the effects of perceived vulnerability and perceived severity factors in the threat appraisal of domestic tourists, the effects of the response efficacy, response cost, and self-efficacy factors in the coping appraisal, and the effects of fear and hope factors as the anticipatory emotion responses regarding protection motivation. Another important aim of the study as to measure the effect of protection motivation on the factors of taking a vacation and avoiding vacation, which constitute behavioral intentions. The third aim was to determine whether protection motivation had a mediating effect between behavioral intentions, threat and coping appraisals, and emotional responses. Moreover, suggesting a model in accordance with the results of these effects is a crucial output of the research. The study aimed to provide suggestions on how the perceptions of local people (as potential domestic tourists) regarding COVID-19 have changed and how the findings can be used to improve the image and marketing strategies of destinations. Tour operators and tourism service providers may target specific groups of potential tourists in their marketing and promotional campaigns. This targeting can be done according to whether people have developed a motivation to protect against a threat, rather than socio–demographic characteristics.

This study sheds light on tourist behavior during a pandemic. Although the impact of the pandemic is lessening, this study contributes to the literature on pandemic tourism, which still requires further data. Second, the study used PMT to clarify the connection between anticipatory emotions and holiday intentions in the context of pandemics. To address this gap, this research aimed to improve the comprehension of preservative holiday behavior during a pandemic by investigating the interaction between anticipatory emotions and protective behavioral intentions in the decision-making process of domestic tourists. Although the protective motivation theory is widely used in tourism and hospitality, it has not been expanded to clarify the relationship between holiday behavior/intentions and anticipatory emotions during a pandemic crisis. Therefore, this study adds to the knowledge layer in this field. Similar to [22], recent PMT studies on tourism have focused on one part of all variables (e.g., perceived vulnerability or perceived severity). However, this study uses a quantitative research approach to examine the emotional reactions and behavioral intentions of residents in Türkiye who have the potential to reduce the harsh impact of the pandemic but have been in an economic crisis for more than a decade.

The results show the factors tourists would like to maintain to assure their health and safety during the holiday. The findings support the main hypotheses of PMT, in which threat and coping appraisals may affect the intention of people to protect themselves from COVID-19 when traveling. Moreover, the study results shed light on how PMT variables (i.e., vulnerability, severity, efficacy, and self-efficacy) are linked to each other. The impact of anticipatory emotions, such as fear and hope, were also considered. Although the hope variable was ineffective in predicting the protection motivation, it was also theoretically significant and requires further investigation. Several studies have shown that perceived severity and vulnerability to contagious diseases positively affect the intention of people to adopt the recommended protective measures (e.g., [26]). The present study results contribute to the literature, which examines the social cognitive processes that govern individuals’ protective behavior and anticipatory emotions.

Significant correlations between PMT variables, behavioral intention, and anticipatory emotions may predict an individual’s protective behaviors toward COVID-19 and the threat of similar threats in the future. In line with the PMT, respondents who disclosed a more severe threat of COVID-19 infection reported greater intention to “regulate all behaviors to avoid the virus” than those who revealed higher rewards for noncompliant protective behaviors. These behaviors included staying at home as much as possible, wearing masks and gloves, using hand sanitizers, and maintaining social distance by avoiding crowds and maintaining a recommended distance from everyone. Participants in this study had high scores on severity, vulnerability, self-efficacy, responsiveness, and fear. Therefore, they revealed a high motivation to engage in protective behaviors. When the risk was not perceived as severe (low perceived severity), individuals were less eager to react and make superficial judgments about the efficiency of warnings. Namely, if society does not understand the threatening remark and its degree of severity, they will neglect the appropriate suggestions.

PMT does not presume that people provide reasonable judgments. A pragmatic judgment about its significance can influence any cognitive evaluation process. Strong notions about severity, vulnerability, self-efficacy, and response efficacy will elicit protection motivation, stop dangerous acts, and produce protective behaviors [75]. Messages that solicit fear may be more efficient in encouraging protective behaviors among those who already perform them [37].

Numerous health studies have shown that fear is positively associated with people’s risk avoidant behavior during or after a disease [9]. When a health outbreak provokes tourists’ fear, they may avoid travel as an immediate protective measure to lessen the probable risks [7]. The effect on travel intentions is the opposite of what risk tolerance would be in the case of avoiding uncertainty and ambiguity about the pandemic and the resulting fear of an even more distant and unfamiliar future. People with the same risk tolerance but a higher tolerance for uncertainty and ambiguity are more likely to travel earlier than people with a higher tolerance for uncertainty and ambiguity [11]. Travel risk studies confirmed that an individual’s protection motivation is an essential indicator of their holiday avoidance [12] or acceptance of protective holiday behaviors [12]. Motivation can significantly influence travel avoidance and holiday intentions of tourists following an outbreak, as resilience can significantly enhance individual adaptive behavior in the face of a threat [14]. Namely, individuals with high psychological strength are more likely to exhibit careful behaviors rather than avoidance behaviors after the pandemic.

Fear can cause individuals to think about the threat more critically, accelerating their motivation to protect themselves [76]. When a pandemic strikes, people’s perceived travel threat can markedly provoke their pandemic travel fear. As a result, individuals are encouraged to take action to protect themselves from traveling after a pandemic. However, a person’s perceived ability to cope with COVID-19 does not lower the level of anxiety, nor does the perceived effectiveness of recommended behaviors to minimize exposure to COVID-19 increase a person’s sense of hope [2]. Fear can make people more critical of a threat and increase their motivation to protect themselves [76]. During a pandemic outbreak, people’s perception of the danger of travel can significantly increase their fear of a pandemic. Therefore, people are encouraged to adopt protective measures against post-pandemic diseases. Although H7, which shows the effect of hope on protection motivation, was accepted, there was no effect of hope on protection motivation. This is due to the fact that participants had higher levels of fear about COVID-19 and lower levels of hope. The averages of the variables also confirm this result.

In conclusion, the results of the research contribute to the existing literature in several ways. First, the study enhances the literature on protective motivation theory and COVID-19-related threat appraisal by providing a framework for the simultaneous role of psychological emotions and personal values through a sequential transmission process. Second, it scrutinizes the relationship between emotions and behavioral intentions by examining the subdimensions of the concepts, their interactions, and their combined impact on tourists’ protective behavior in the post-COVID era through the lens of protection motivation theory.

## 6. Theoretical and Practical Implications

In a tourism context, the results contribute to the comprehension of tourists’ health behaviors. Rather than using PMT only to describe tourists’ propensity to stay away from risky circumstances, the full PMT model has been tested, and its influence on tourists’ health intentions and behaviors has been explained. In terms of tourism and the travel context, the results highlight that preventive measures against health risks are popular among national tourists, such as those looking for health information before traveling. In addition to tourism, this study advances the theory validation [77]. Wit the use of PMT deductive reasoning and related literature, nine hypotheses were developed and tested in the study. The results confirmed eight hypotheses, and it was observed that the response cost did not affect protection motivation. The reason for this is that the model predicts that the higher the response efficacy, self-efficacy, and the lower response cost, the more one will decide to perform adaptive behaviors [78]. Furthermore, individuals will be more likely to adopt protective behaviors if the associated response costs are low. Another explanation for this could be that response costs have primarily aligned with the acceptance of social distancing rules and easy access to any kind of protective tools, such as masks and disinfectants. In this way, the participants have already adopted protective behaviors in response to the pandemic threat. These results warrant further experimentation on this topic.

From a practical point of view, the sub-constructs of PMT were significantly associated with protective behavioral intention, which proposes that future interventions should focus on people’s perceptions. This result also contributes to developing adequate scientific evidence of COVID-19 severity and vulnerability and enhances response efficacy and self-efficacy through skills training.

In terms of practical implications, this study contributes to existing knowledge about how to stimulate tourists to be involved in risk prevention and mitigation. Assuring the health and safety of individuals is naturally encouraging for them, and researchers and practitioners help in the achievement of this goal somehow. Second, the results indicate that more attention needs to be paid to the key PMT pathway to achieve the desired behavioral changes in potential tourists. Future endeavors should aim to strengthen tourists’ perceived efficacy (both self-efficacy and response efficacy) [79]. To this end, campaigns and communication strategies should be provided to highlight the precautionary measures, and they should be all accessible to the individuals. Tourism stakeholders such as tour operators, travel agencies, and hotel and destination managers should reinforce marketing strategies, advertising, and promotion campaigns. Further, tourism stakeholders should comply with the policies and guidelines of service providers to enhance safety and should conduct promotional activities to attract tourists. Tourists will benefit from this study through increased awareness of the risk that accompanies an infectious disease, such as COVID-19. They need to understand how a virus spreads and the importance of protective behaviors in keeping themselves healthy. Inexperienced tourists need to be educated about the pandemic as well. It is crucial to pay attention to the fact that tourists play a key role in preventing the spread of a pandemic during holidays by practicing healthy behaviors.

## 7. Limitations and Future Directions

Certain limitations may limit the generalizability of the findings. First, the study was carried out with national respondents in Türkiye. Therefore, the results cannot be extrapolated to international tourists outside of Türkiye. Another limitation is the non-random sample of respondents. Therefore, conducting randomized research is a suggested extension of this study. Reiteration of the research in different regions and countries, considering regional infection risks and other sociodemographic variables that may raise an individual’s vulnerability, such as his or her income, cultural values, and personality, could be considered another valuable extension of this study. One of the limitations of the research population is that the participants had at least one holiday experience in the last five years.

Data for this study were also collected in the final months of the pandemic in Türkiye when the restrictions started to loosen, the infected cases were no longer as high as in the previous stages, and there were no lockdowns. For this reason, related future research could focus on long-term investigations in different time series of the pandemic to observe whether public trust in government and related factors that influence the continuity of protection-seeking behavior have changed. For the overall research model, it is crucial to consider a few additional relationships. Since the research model needs to be kept parsimonious, it should be mentioned that some of the relationships were not included intentionally in the study (e.g., direct relationships between some of the factors that were included as exogenous variables, such as severity and vulnerability influencing fear). Furthermore, the demographics of the sample consisted of a very high share of academics, which is another potential limitation of the research.

Tourists’ intention to visit a destination in the future may depend on the extent to which they believe that the destination is acting as an extrinsic motivator by addressing the health, social, and economic consequences of the disease and attempting to contain the spread of the disease locally and globally. Due to the complementary features of the theory of planned behavior (TPB) and PMT [80], an integrated model with TPB and PMT will provide a more comprehensive understanding of the complexity involved in the sustainable behavioral intentions in the post-COVID era.

## Figures and Tables

**Figure 1 healthcare-10-01623-f001:**
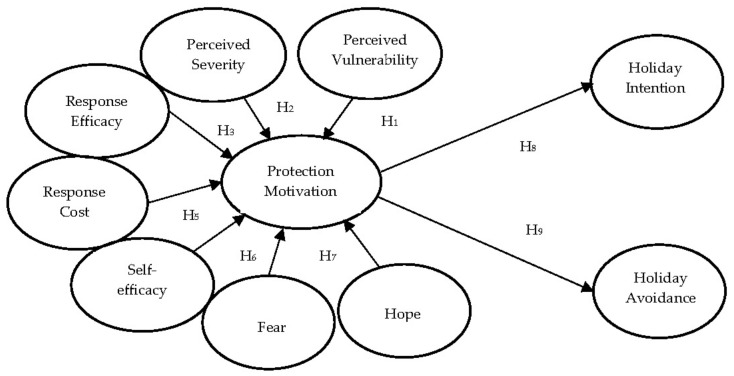
Research model.

**Figure 2 healthcare-10-01623-f002:**
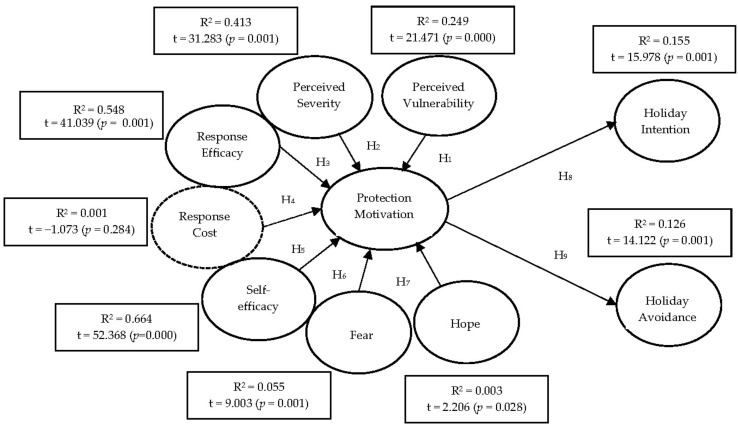
Estimated structural model.

**Table 1 healthcare-10-01623-t001:** Items and constructs.

**Perceived Vulnerability (PV)**	**Sources**
I will be easily infected with COVID-19 if I travel	Qiao et al. [31]
Social distancing is vital when traveling during a COVID-19 outbreak	
Traveling is scary while COVID-19 transmission persists	
**Perceived Severity (PS)**	
COVID-19 is highly contagious	Qiao et al. [31]
COVID-19 has widespread community transmission	
COVID-19 is a serious threat to human life	
**Response Efficacy** **(RE)**	
Efforts to keep safe from COVID-19 threats are effective	Zheng et al. [43]
Available measures to protect myself from being infected by COVID-19 are effective	
Measures that can be taken to stop tourists from being infected by COVID-19 are adequate	
Preventive measures to stop tourists from being infected by COVID-19 are adequate	
I am less likely to be exposed to the COVID-19 threat if the preventive measures are performed	
**Response Cost (RC)**	
The price of disinfectants is high	Rad et al. [60]
It is hard for me to find a mask	
Disinfecting objects and places suspected of COVID-19 is time-consuming	
It is hard for me to stay at home and not be in crowded places	
Washing hands frequently with soap and water can hurt the skin of my hands	
I am ashamed of not shaking hands with my friends	
I am allergic to the smell of disinfectants	
**Self-efficacy (SE)**	
Taking measures to prevent COVID-19 infection is easy	Zheng et al. [43]
I have the necessary skills and equipment to protect myself from being infected by COVID-19	
My skills and the equipment required to stop COVID-19 infecton are adequate	
I could learn to perform preventive measures to protect myself from being infected by COVID-19	
**Fear (F)**	
When thinking about COVID-19, to what extent do you feel frightened	Zheng et al. [43]
When thinking about COVID-19, to what extent do you feel nervous	
When thinking about COVID-19, to what extent do you feel anxious	
**Hope (H)**	
When thinking about COVID-19, to what extent do you feel hopeful	Kim et al. [2]
When thinking about COVID-19, to what extent do you feel optimistic	
When thinking about COVID-19, to what extent do you feel encouraged	
**Protection Motivation (PM)**	
I protect myself from being infected by COVID-19 when traveling	Zheng et al. [43]
I engage in activities that protect myself from being infected by COVID-19	
I expend effort to protect myself from being infected by COVID-19	
I obey policies to protect myself from being infected by COVID-19	
**Holiday Intention (HI)**	
COVID-19 will affect my decision whether to go on holiday in 2022	Pappas [51]
COVID-19 will affect my decision whether to go on holiday in future years	
Due to COVID-19, I would prefer to go on holiday somewhere in Türkiye rather than abroad	
COVID-19 has had a greater impact upon my holiday intention than the recession	
I intend to go on holiday during 2022	
**Holiday Avoidance (HA)**	
I will be avoiding traveling abroad for at least one year	Turnšek et al. [61]
I will prefer to stay home this summer as a precaution and not go on any vacations	
If traveling, I will avoid public transport	
In the future, I will no longer attend crowded events due to the fear of COVID-19	
I will have no problem with using planes, buses or trains as they will be safe again soon	
This year I would rather look for holiday possibilities within my own country	
Once the problems are over, I will travel extensively to make up for lost time	

**Table 2 healthcare-10-01623-t002:** Profile of respondents.

Sample (n = 1391)					
Characteristics	Frequency	%	Characteristics	Frequency	%
**Gender**			**Occupation**		
Female	755	54.3	Public sector	395	28.4
Male	636	45.7	Private sector	589	42.3
**Age**			Housewife	105	7.5
18–25	243	17.5	Other	302	21.7
26–33	354	25.4	**The number of holidays in the last five years**		
34–41	423	30.4	1 time	439	31.6
42–49	189	13.6	2 times	377	27.1
50–57	144	10.4	3 times	309	22.2
58 and above	38	2.7	More than 4	266	19.1
**Education**			**Have you ever been COVID-19 positive?**		
Primary school	60	4.3	Yes	477	34.3
High School	482	34.7	No	914	65.7
College/University	811	58.3	**Have you had a COVID-19 vaccine?**		
Higher than graduate	38	2.7	Yes	1147	82.5
**Level of monthly income**			No	244	17.5
Below 2800 ₺	213	15.3			
2801 ₺–5000 ₺	436	31.3			
5001 ₺–10.000 ₺	495	35.6			
10.001 ₺ and above	247	17.8			

**Table 3 healthcare-10-01623-t003:** Mean, standard deviation, skewness coefficient, and factor values of items.

Items	X^−^	SD	Skewness– Kurtosis	Load
**Perceived Vulnerability**				
I will be easily infected with COVID-19 if I travel	2.59	1.119	0.267/−0.662	0.81
Social distancing is vital when traveling during a COVID-19 outbreak	3.79	1.184	−0.891/−0.064	0.52
Traveling is scary while COVID-19 transmission persists	2.82	1.165	0.073/−0.86	0.87
**Perceived Severity**				
COVID-19 is highly contagious	3.62	1.136	−0.709/−0.157	0.89
COVID-19 has widespread community transmission	3.64	1.151	−0.708/−0.213	0.93
COVID-19 is a serious threat to human life	3.64	1.142	−0.669/−0.276	0.82
**Response Efficacy**				
Efforts to keep safe from COVID-19 threats are effective	3.41	1.093	−0.436/−0.350	0.79
Available measures to protect myself from being infected by COVID-19 are effective	3.62	1.100	−0.677/−0.087	0.80
I am less likely to be exposed to the COVID-19 threat if preventive measures are performed	3.38	1.101	−0.423/−0.421	0.69
**Response Cost**				
It is hard for me to find a mask	2.49	1.202	0.405/−0.757	0.66
I am ashamed of not shaking hands with my friends	2.39	1.192	0.465/−0.731	0.72
I am allergic to the smell of disinfectants	2.41	1.179	0.448/−0.759	0.75
**Self-efficacy**				
Taking measures to prevent COVID-19 infection is easy	3.29	1.089	−0.308/−0.497	0.70
I have the necessary skills and equipment to protect myself from being infected by COVID-19	3.45	1.097	−0.402/−0.418	0.88
My skills and the equipment required to stop from being infected by COVID-19 are adequate	3.24	1.106	−0.280/−0.544	0.73
I could learn to perform preventive measures to protect myself from being infected by COVID-19	3.57	1.074	−0.690/−0.016	0.79
**Fear**				
When thinking about COVID-19, to what extent do you feel frightened	2.41	1.133	0.205/−0.535	0.83
When thinking about COVID-19, to what extent do you feel nervous	2.48	1.142	0.190/−0.531	0.91
When thinking about COVID-19, to what extent do you feel anxious	2.62	1.199	0.173/−0.571	0.68
**Hope**				
When thinking about COVID-19, to what extent do you feel hopeful	2.66	1.097	0.367/−0.657	0.89
When thinking about COVID-19, to what extent do you feel optimistic	2.69	1.111	0.349/−0.570	0.94
When thinking about COVID-19, to what extent do you feel encouraged	2.41	1.133	0.236/−0.765	0.89
**Protection Motivation**				
I protect myself from being infected by COVID-19 when traveling	3.48	1.093	−0.519/−0.301	0.72
I engage in activities that protect myself from being infected by COVID-19	3.44	1.082	−0.455/−0.348	0.84
I expend effort to protect myself from being infected by COVID-19	3.58	1.089	−0.643/−0.149	0.91
I obey policies to protect myself from being infected by COVID-19	3.61	1.086	−0.681/−0.050	0.83
**Holiday Intention**				
COVID-19 will affect my decision whether to go on holiday in 2022	2.95	1.216	−0.091/−0.887	0.82
COVID-19 will affect my decision whether to go on holiday in future years	2.93	1.204	−0.089/−0.874	0.96
Due to COVID-19, I would prefer to go on holiday somewhere in Türkiye rather than abroad	3.13	1.220	−0.273/−0.814	0.68
COVID-19 has had a greater impact upon my holiday intention than the recession	2.78	1.292	0.123/−1.043	0.56
**Holiday Avoidance**				
I will be avoiding traveling abroad for at least one year	2.99	1.271	−0.070/−0.937	0.73
I prefer to stay home this summer as a precaution and not go on any vacations	2.81	1.215	0.102/−0.843	0.80
If traveling, I will avoid public transport	2.86	1.187	0.075/−0.824	0.71
In the future, I will no longer attend crowded events due to the fear of COVID-19	2.94	1.151	−0.024/−0.720	0.74

**Table 4 healthcare-10-01623-t004:** Evaluation range of arithmetic means.

Evaluation Range	Likert Value		Range Value
4.21–5.00	Strongly agree	Extremely	Quite
3.41–4.20	Agree	Very	Positive
2.61–3.40	Neither agree nor disagree	Moderately	Medium
1.81–2.60	Disagree	Slightly	Negative
1.00–1.80	Strongly disagree	Not at all	Quite

**Table 5 healthcare-10-01623-t005:** Results of the evaluation form of the participants.

Factor	X^−^ ± SS	Max–Min	Range Value
**PV**	3.0657 ± 0.88047	3.79–2.59	Medium
**PS**	3.6336 ± 1.02815	3.64–3.62	Positive
**RE**	3.4682 ± 0.90593	3.62–3.38	Positive
**RC**	2.4299 ± 0.93993	2.49–2.39	Negative
**SE**	3.3907 ± 0.88341	3.57–3.24	Medium
**F**	2.6715 ± 1.01938	2.62–2.41	Medium
**H**	2.5037 ± 0.98370	2.69–2.41	Negative
**PM**	3.5286 ± 0.91704	3.61–3.44	Positive
**HI**	2.9466 ± 0.97106	3.13–2.78	Medium
**HA**	2.8990 ± 0.95743	2.99–2.81	Medium

**Table 6 healthcare-10-01623-t006:** Composite reliability, average variance extracted, and square root of AVE.

Factors	CA	CR	AVE	1	2	3	4	5	6	7	8	9	10
**PV**	0.64	0.79	0.56	**0.748**									
**PS**	0.88	0.91	0.78	0.362	**0.883**								
**RE**	0.77	0.81	0.58	0.267	0.393	**0.762**							
**RC**	0.70	0.75	0.51	0.028	0.000	0.000	**0.714**						
**SE**	0.82	0.86	0.61	0.188	0.320	0.497	0.002	**0.781**					
**F**	0.91	0.93	0.82	0.133	0.094	0.046	0.016	0.039	**0.906**				
**H**	0.81	0.85	0.66	0.001	0.002	0.007	0.026	0.012	0.054	**0.812**			
**PM**	0.87	0.90	0.69	0.249	0.414	0.548	0.001	0.664	0.055	0.004	**0.831**		
**HI**	0.80	0.85	0.59	0.221	0.126	0.147	0.104	0.156	0.105	0.009	0.155	**0.768**	
**HA**	0.81	0.83	0.56	0.254	0.112	0.142	0.148	0.132	0.094	0.010	0.125	0.468	**0.748**

**Table 7 healthcare-10-01623-t007:** HTMT results.

**PV**										
**PS**	0.831									
**RE**	0.768	0.762								
**RC**	0.224	0.003	0.008							
**SE**	0.625	0.648	0.886	0.064						
**F**	0.474	0.342	0.258	0.160	0.228					
**H**	0.042	0.049	0.111	0.214	0.138	0.267				
**PM**	0.701	0.736	0.910	0.038	0.964	0.263	0.072			
**HI**	0.651	0.423	0.490	0.436	0.491	0.378	0.118	0.473		
**HA**	0.693	0.397	0.481	0.508	0.449	0.367	0.122	0.425	0.856	
	**PV**	**PS**	**RE**	**RC**	**SE**	**F**	**H**	**PM**	**HI**	**HA**

**Table 8 healthcare-10-01623-t008:** Estimated model evaluation.

Hypothesis		*f* ^2^		R^2^	β	t	*p*	SE	H
**H1**	**PV→PM**	0.33	Medium Effect	0.249	0.499	21.471	0.000	0.023	**S**
**H2**	**PS→PM**	0.70	Large Effect	0.413	0.643	31.283	0.001	0.021	**S**
**H3**	**RE→PM**	1.21	Large Effect	0.548	0.740	41.039	0.001	0.018	**S**
**H4**	**RC→PM**	0.00	No Effect	0.001	−0.029	−1.073	0.284	0.027	**N**
**H5**	**SE→PM**	1.98	Large Effect	0.664	0.815	52.368	0.000	0.016	**S**
**H6**	**F→PM**	0.06	Small Effect	0.055	0.235	9.003	0.001	0.026	**S**
**H7**	**H→PM**	0.00	No Effect	0.003	0.059	2.206	0.028	0.027	**S**
**H8**	**PM→HI**	0.18	Medium Effect	0.155	0.394	15.978	0.001	0.025	**S**
**H9**	**PM→HA**	0.14	Small Effect	0.126	0.354	14.122	0.001	0.025	**S**

**Table 9 healthcare-10-01623-t009:** Mediating effect of protection motivation (PM).

Factors	Total Effect					Direct Effect			Indirect Effect		
	*f* ^2^	R^2^	β	SE	t	β	SE	t	β	LLCI	ULCI
**PV→HI**	0.284	0.221	0.470	0.024	19.8606 **	0.364	0.027	13.6282 **	0.106	0.0751	0.1373
**PS→HI**	0.144	0.126	0.355	0.025	14.1485 **	0.173	0.031	5.4329 **	0.182	0.1365	0.2294
**RE→HI**	0.172	0.147	0.383	0.025	15.4390 **	0.201	0.036	5.5484 **	0.181	0.1253	0.2365
**RC→HI**	0.116	0.104	0.322	0.025	12.6624 **	0.333	0.023	14.4905 **	−0.012	−0.0368	0.0130
**SE→HI**	0.185	0.156	0.395	0.025	16.0213 **	0.220	0.042	5.2168 **	0.175	0.1067	0.2444
**F→HI**	0.117	0.105	0.324	0.025	12.7794 **	0.245	0.024	10.0104 **	0.079	0.0564	0.1024
**H→HI**	0.008	0.008	0.092	0.027	3.4358 **	0.069	0.025	2.7900 *	0.023	−0.0006	0.0469
**PV→HA**	0.340	0.254	0.504	0.023	21.7469 **	0.436	0.027	16.4388 **	0.068	0.0391	0.0985
**PS→HA**	0.125	0.111	0.334	0.025	13.1978 **	0.180	0.032	5.5749 **	0.153	0.1067	0.2026
**RE→HA**	0.166	0.142	0.377	0.025	15.1709 **	0.254	0.037	6.9137 **	0.123	0.0666	0.1800
**RC→HA**	0.172	0.147	0.384	0.035	15.4816 **	0.394	0.023	17.3072 **	−0.011	−0.0346	0.0127
**SE→HA**	0.152	0.132	0.363	0.025	14.5298 **	0.222	0.043	5.1706 **	0.142	0.0723	0.2076
**F→HA**	0.103	0.093	0.306	0.026	11.9600 **	0.235	0.025	9.3991 **	0.070	0.0490	0.0924
**H→HA**	0.010	0.010	0.099	0.027	3.7186 **	0.079	0.025	3.1379 *	0.021	−0.0002	0.0434

** *p* < 0.01, * *p* < 0.05.

## Data Availability

Data available on request.

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
