# Peer review of "Understanding the Behavioral Intentions about Holidays in the Shadow of the COVID-19 Pandemic: Application of Protection Motivation Theory"

_healthcare, 2022, doi:10.3390/healthcare10091623_

Round 1
Reviewer 1 Report (New Reviewer)
The authors use advanced methods of psychometrics and structural equation modelling (SEM) to elicit emotional response and protective behaviors and holiday intentions or holiday avoidance by the Turkish domestic tourists. The sample size (N = 1391) is large enough to provide fir meaningful SEM computations.
The theoretical background of the paper refers to the protection motivation theory (PMT). This is OK for purely psychological determinants of the tourist behaviour. The theoretical part of the paper, unfortunately, largely misses reference to theory on perception of risk and uncertainty in tourism. There are both general and domain (tourism)-specific perceptions of risk and uncertainty (see Quintal et al 2010, Williams and Balaz 2015. Yang et al 2017). There also is literature on Covid-related tourist behaviour (Arbalú et al 2020: Williams et al 2022).
The authors also should provide detailed information about theoretical resources for specific constructs in their questionnaire (item-by-item).
The authors may consider (1) re-working their literature review: (2) integrate incorporate tourism-related insights on risk & uncertainty perceptions and behaviours; (3) re-interpret their conclusions accordingly. Are there, for example, any implications for tourism policies and management? It also is unclear, how the conclusions are limited by local context of the research.
Author Response
First of all, we would like to thank the reviewers for their valuable feedback and suggestions. We have tried to improve the manuscript based on their valuable feedback. Our answers to their questions:
- The scales of the variables were taken from the studies as a whole. Therefore, an item-by-item explanation can not be made. The studies from which the scales were taken are shown in Table 1.
- The resources you suggested have been reviewed and insights have been added on related topics by line 59, 85, 611 and 678.

Reviewer 2 Report (New Reviewer)
The article deals with a very relevant topic in current research, especially in the context of the international debate on the protection behaviour of tourists in the post-pandemic period. The article focuses on the recent scientific literature on the subject, highlighting the advances and contributions of various authors. It is a well-argued and well-founded article. It is well written and structured, and easy to read. However, there are areas for improvement;
1. Figure 2 could depict the outcome ratios to make it easier to understand.
2. It is stated that the findings can be used to improve the image and marketing strategies of destinations (line 98 and 523) but it does not state how.
3. In the practical implications, campaigns and communication strategies to highlight the precautionary measures are proposed but specific measures are not developed.
Author Response
First of all, we would like to thank the reviewers for their valuable feedback and suggestions. We have tried to improve the manuscript based on their valuable feedback. Our answers to their questions:
1. Figure 2 could depict the outcome ratios to make it easier to understand. We added the outcome ratios on Figure 2.
2. It is stated that the findings can be used to improve the image and marketing strategies of destinations (line 98 and 523) but it does not state how. We added related explanations by line 555.
3. In the practical implications, campaigns and communication strategies to highlight the precautionary measures are proposed but specific measures are not developed. We added related explanations by line 678.

Reviewer 3 Report (New Reviewer)
The text presents an interesting research. In general, the structure of this manuscript is complete and comprehensive. I suggest a minor revision with comments that should be emphasized as follows:
- The abstract needs to be condensed.
- As for the Protection Motivation Theory, the application reason as well as its importance could be enriched.
- As for the variables fear and hope, I suggest that the authors may emphasize more to strengthen the representativeness of the anticipatory emotion responses.
- As for the data collection part, the study includes residents who have had at least one-holiday experience in the last five years and the survey was conducted online. Why 5 years and why not on-site survey? Authors are expected to explain more.
- Theoretical contributions and practical implications can be polished further.
Author Response
First of all, we would like to thank the reviewers for their valuable feedback and suggestions. We have tried to improve the manuscript based on their valuable feedback. Our answers to their questions:
- The abstract needs to be condensed. “Moreover, understanding individuals’ protection motivations regarding the pandemic is vital for assistance in the review of the tourism industry after the outbreak.” This part is deleted from the abstract.
- As for the Protection Motivation Theory, the application reason as well as its importance could be enriched. We added related information by line 90.
- As for the variables fear and hope, I suggest that the authors may emphasize more to strengthen the representativeness of the anticipatory emotion responses. We added related information by line 217.
- As for the data collection part, the study includes residents who have had at least one-holiday experience in the last five years and the survey was conducted online. Why 5 years and why not on-site survey? Authors are expected to explain more. We added related information by line 355, 358 and 697.
- Theoretical contributions and practical implications can be polished further. We added related information by line 678.

Round 2
Reviewer 1 Report (New Reviewer)
Most comments were processed in an appropriate way. The paper was improved. I recommend the paper for publication.
This manuscript is a resubmission of an earlier submission. The following is a list of the peer review reports and author responses from that submission.
Round 1
Reviewer 1 Report
I have read the manuscript, and I have some suggestions for the authors:
1. The gaps identified in the manuscript do not seem to be significant. Therefore, it should be bolstered.
2. Please provide the Bootstrap results in Table 8
3. Please provide an in-depth discussion that why response cost does not affect protection motivation. Likewise, the H8 and H9 hypotheses should be also discussed in-depth.
4. The practical implication section is weak and inconsistent with the findings.
Reviewer 2 Report
The article is interesting for researchers from different disciplines. You can see that the authors have put a lot of work in its preparation. The article enriches our knowledge about motivation, tourists' behavior in the face of crisis situations.
Reviewer 3 Report
I want to thank the authors for this interesting investigation of the effect of protection motivation and some of its antecedents on the intention of individuals to travel again in the aftermath of the pandemic. Overall, I consider the topic to be interesting and relevant, and the applied methodology is for most parts rigorous (including a large survey sample). There are still some areas that should be addressed though related to (i) conceptualization and structure of the paper, and (ii) the methodology. I want to highlight the main issues in these areas and then provide some additional minor comments. In most cases I will highlight the issues within these categories in the order in which they appear in the manuscript.
Conceptualization and Structure. On p. 2 the authors argue that “[t]here is little empirical evidence on the link between PMT and international travel” yet they focus on domestic tourism (they state as much on the same page, line 84). I would avoid recommend avoiding such arguments that are not directly related to the topic of the paper. On p. 2, line 82 the sentence that addresses whether the article falls within the scope of the journal should be removed (this should be part of a side letter to the editors and not part of the body of the paper). On p. 2-3 (last / first paragraph) the authors summarize their paper. I would suggest avoiding a summary of this kind in the introduction as we already have a summary in the abstract, and the concluding part of the paper. Instead, I would suggest that the authors briefly mention the research question they want to answer and then highlight the structure of their paper.
A main issue of section 2 is its current structure. In particular in section 2.1, the authors mention many variables and relationships between them, sometimes seemingly at random. For example, threat appraisal and coping appraisal are mentioned initially, but only explained sporadically in later sections. I would suggest that the authors initially present their selected theory (here only focus on relevant parts, e.g. “multilevel models” are mentioned on p. 3, but then not explained – this should be avoided), then their main construct (i.e., protection motivation), then the two types of appraisal process that could initiative protection motivation and finally the variables that fall within each of these two processes. Ideally, the authors should also include a table that includes an overview of their constructs and their definition.
For the formulated hypotheses I would also suggest some adaptations:
H1-5 should not include reference to holiday intentions otherwise they suggest that a mediation is tested as part of this hypothesis (i.e., with protection motivation as mediator)
H6-9 should be directed as the relationships under investigation are not exploratory (i.e., is a positive / negative effect expected).
For the overall research model, I would suggest considering some additional relationships. While I understand that the authors want to keep their research model parsimonious, it should be mentioned (at least as part of the Limitations) that some of the relationships mentioned on p. 5 and 6 were not included intentionally (e.g., direct relationships between some of the factors that were included as exogenous variables such as severity and vulnerability influencing fear). What would be a meaningful extension though is a test of the role of protection motivation as a mediating variable (e.g., as mentioned by the authors on p. 6). For this purpose also direct relationships between the 7 independent variables and the two outcome variables should be included in a tested model and the indirect effects through protection motivation should be reported.
Methodology. On p. 6 (and at other points of the paper) the authors claim that they tested their model “experimentally”. As there was no manipulation included, I would suggest avoiding this impression.
On p. 8 for the sampling approach the authors mention a combination of convenience sampling and snowballing. As snowballing was involved I was wondering how the authors know that 2,500 individuals were reached by an invitation or whether they meant that 2,500 individuals started the survey. In addition, I would suggest that the authors add further information on how the survey was distributed (e.g., did the authors post it on social media channels; was it provided through a mailing list of their university, etc.). For the sampling it should also be mentioned which indicators were used to decide whether a completed survey should be excluded (n = 156) or not. On p. 9, the authors include the demographics of their sample, which shows a very high share of academics. This should be briefly discussed as a potential limitation as it is highly unlikely that in the general population of Turkey which has travelled within the last five years, almost 60% hold an academic degree.
On p. 9 the authors do not provide any detail on the model testing procedures. Which type of regression was used or did the authors decide to use structural equation modeling (which I would highly recommend). If covariance-based SEM was conducted in LISREL, then goodness of fit indicators need to be reported for the CFA and the structural model test (fit values are also mentioned on p. 10 as a motivation to exclude items, which gives the impression that SEM was applied).
For Table 1 I would suggest a more meaningful title such as “Items and Constructs” and also the authors should already highlight which items are later on not included in the structural model tests (i.e., highlight the 10 items that are mentioned on p. 10). In addition, it is not meaningful to include the Cronbach’s Alpha for the initial scale, if it is the not used for the actual structural model test. Rather, I would suggest that the authors include the reliability scores of the final scales. If these are the “CA” values in Table 6 (an assumption of mine), then those values should also be provided here. Further, the authors need to clarify why CA values below .70 are accepted (perceived vulnerability has a score of .64).
On p. 11, in Table 4, I would suggest that “Quite positive” and “positive” as well as “Quite negative” and “negative” are switched as “Quite…” expresses that it is not fully positive / negative. Table 5 then also has to be adapted accordingly.
On p. 11, after Table 5, the authors should start with a new section “4.3 Measurement Model Evaluation” to better distinguish between these main parts of the analysis.
On p. 13, for the main results I am missing the overall R² score for protection motivation. Here, I would then also suggest that the authors add the investigation of a mediating effect for all 7 independent variables through protection motivation, which would further strengthen the contribution as it would demonstrate the value of this specific variable.
Minor. The authors need to implement a rigorous round of proofreading. Here are just some examples of cases which need to be resolved and for some of them (highlighted with [?]) I am not even sure how to resolve them (list is non-exhaustive; problematic areas and/or suggestions for improvement are in brackets):
p. 2: “Most [studies] about post-pandemic holiday behavior…”
p. 2: “…(PMT) has received [] scientific attention…”
p. 2: “…this study will help [us to understand] how the perceptions of domestic tourists…”
p. 4: “[Threat] [?] appraisal measures maladaptive responses … whereas [threat] [?] appraisal…”
p. 4: “The classical theory of primary emotion [proposed] by Ekman…”
p. 5: “Sometimes people fear or hope that something undesirable may [/may not ] happen…”
p. 5: “Fear is expressed as an [informative communication] [?] about a threat…”
p. 7: “…to measure the [constructs] that make up the model…”
p. 8: authors use “universe” but likely meant “population”
p. 9: “…to determine the [] validity and reliability…”
p. 9, Table 2: “Have you ever been COVID-19 [positive]?”
p. 10: “expressions” should most likely be replaced by “statements” or “items”
In addition, also the style of writing throughout the paper should be on the same level. For example, Section 2.2 has a very different style (wording, sentence structure, use of references) than the other parts of the manuscript.
Overall, an interesting study with potential, but still some work that has to be done. I hope that the authors can profit from my comments and wish them all the best for their future research.
Reviewer 4 Report
The proposed manuscript is of interest to the tourism industry as it offers results, conclusions and recommendations that are beneficial to the tourism business as well as to planning institutions.
In my opinion, however, it would only benefit if some additions and adjustments were made. They are mostly of a structural and technical nature. And they refer to the following:
- it would be good to make the abstract more concise as an exposition, without the use of abbreviations (it is enough to present the abbreviations in the Introduction, and then to use only in this abbreviated form)
- it is not necessary to present the individual hypotheses in the abstract, it is enough to draw concrete conclusions
- the sources finally mentioned are 93 in number; this is definitely a lot; specifying their number could be considered
- to specify the citation - in some places it is good to indicate the names of the authors (eg line 72, source 25) or when first quoting an author team to write all of them
- to reconsider the derivation of hypotheses within the scope of Literature review Part - As far as I understand, the authors refer to the literature in their hypotheses. It seems to me that a more detailed justification of the conceptual framework in relation to the construction and logic of the study would be needed here
- to refine the model - perhaps it would be better to put it entirely in the part with the methodology. Now fig. 1 is in the literature section, and fig. 2 in the results. My suggestion is to consider their presentation, as well as to write the relevant compilers or sources under the figures and tables
- to refine the discussion by giving more specificity to the application of the results and the different points of view regarding them
- it would be good to have a conclusion that suggests the relationship between the study, the results and the title of the article
In general, with minor stylistic, structural and technical adjustments, the manuscript could improve its quality and be moved to publication.
Round 2
Reviewer 3 Report
I want to thank the authors for the effort they put into the revision. All my comments have been sufficiently addressed and I wish the authors all the best for their future research.